# Risk factors of reattempt among suicide attempters in South Korea: A nationwide retrospective cohort study

**Min Ji Kim**[1,2], **Jeong Hun Yang**[1,2], **Min Jung Koh**[3], **Youngdoe Kim**[3], **Bolam Lee**[3], **Yong Min Ahn**[1,2]*

1 Department of Psychiatry, Seoul National University College of Medicine, Seoul, Korea, 2 Department of Neuropsychiatry, Seoul National University Hospital, Seoul, Korea, 3 Medical Affairs, Janssen Korea Ltd., Seoul, Korea

* aym@snu.ac.kr

**Data Availability Statement:** Data cannot be shared publicly because it is a third party data. Data can be obtained via website of HIRA(HIRA bigdata open portal) by filling out the application. The data

## Abstract

This study aimed to identify underlying demographic and clinical characteristics among individuals who had previously attempted suicide, utilizing the comprehensive Health Insurance Review and Assessment Service (HIRA) database. Data of patients aged 18 and above who had attempted suicide between January 1 and December 31, 2014, recorded in HIRA, were extracted. The index date was identified when a suicide attempt was made within the year 2014. The medical history of the three years before the index date and seven years of follow-up data after the index date were analyzed. Kaplan-Meier estimate was used to infer reattempt of the suicide attempters, and Cox-proportional hazard analysis was used to investigate risk factors associated with suicide reattempts. A total of 17,026 suicide attempters were identified, of which 1,853 (10.9%) reattempted suicide; 4,925 (28.9%) patients had been diagnosed with depressive disorder. Of the reattempters, 391 (21.1%) demonstrated a history of suicide attempts in the three years before the index date, and the mean number of prior attempts was higher compared to that of the non-reattempters (1.7 vs.1.3, $p$-value < 0.01). Prior psychiatric medication, polypharmacy, and an increase in the number of psychotropics were associated with suicide reattempt in overall suicide attempters. (Hazard ratio (HR) = 3.20, 95% confidence interval [CI] = 2.56–4.00; HR = 2.42, 95% CI = 1.87–3.14; HR = 19.66, 95% CI = 15.22–25.39 respectively). The risk of reattempt decreased in individuals receiving antidepressant prescriptions compared to those unmedicated, showing a reduction of 78% when prescribed by non-psychiatrists and 89% when prescribed by psychiatrists. Similar risk factors for suicide reattempts were observed in the depressive disorder subgroup, but the median time to reattempt was shorter (556.5 days) for this group compared to that for the overall attempters (578 days). Various risk factors including demographics, clinical characteristics, and medications should be considered to prevent suicide reattempts among suicide attempters, and patients with depressive disorder should be monitored more closely.

is provided in a DVD (text file) format and a fee for the data is subject to be charged. More detailed information on how to access the database is in the following website: https://opendata.hira.or.kr/op/opc/selectOpenDataAplInfoView.do. Other researcheres would have the same access to these data as the authors, and the authors did not possess any special access privileges not available to others.

**Funding:** This study was funded by Janssen Korea Ltd. The funder had no role in study design, data collection and analysis, decision to publish, or preparation of the manuscript.

**Competing interests:** I have read the journal's policy and the authors of this manuscript have the following competing interests: MJ Koh, Y Kim, and B Lee are employees of Janssen Korea Ltd. YM Ahn declared a research fund from Janssen Korea Ltd. and participated in speakers' events in Janssen Korea Ltd, Lundbeck Korea Co., Ltd., and Korea Otsuka Pharmaceutical. The remaining authors have nothing to disclose.

## Introduction

Suicide is one of the most serious public health concerns, with approximately 703,000 people worldwide dying by suicide each year, according to the World Health Organization [1]. The grave problem is not only a tragedy that affects families and communities but also a leading cause of death among young adults, resulting in a high national economic burden [2, 3]. To prevent suicide, several studies have investigated its risk factors, including sociodemographic factors, clinical diagnoses, and medications [4, 5]. Among the many risk factors that have been studied, a history of suicide attempts is the most critical and has been supported by several studies [6, 7]. Consequently, it becomes necessary to prioritize prevention efforts at the secondary level, targeting individuals who have previously attempted suicide [8]. From this perspective, it is crucial to determine the appropriate duration for the vigilant monitoring of patients and examine the distinct factors influencing reattempts through other means to prevent a potential suicide attempt. According to previous cohort studies on suicide attempters, the highest risk for completed suicide or subsequent suicide attempts occurred within the first two years following the initial attempt. Appropriate support and intervention during this period can play a pivotal role in reducing the risk of further tragic outcomes [7, 9].

Depression has been highlighted as an important risk factor for suicide since it is the most common psychiatric disorder in people who die by suicide [10–12]. Previous studies have confirmed that about 30% of patients with major depressive disorder (MDD) attempt suicide during their lifetime [13]. Other studies have reiterated that individuals diagnosed with depression showed increased rates of suicide mortality and, thus, require close observation across different age groups and cultural backgrounds [14–16]. In this context, the United States Preventive Services Task Force recommended that suicide-risk assessments be based on depression screenings [17]. Early screening and optimizing treatment are important, especially for those who suffer from MDD, because early intervention could lead to the prevention of suicide [18].

Although it has been postulated that psychotropic medications may be associated with suicide attempts, not much evidence has been accumulated to demonstrate how the number of and reasons for prescribing medications affects suicide attempts. Antidepressants are the most frequently prescribed medications for individuals who have attempted suicide; they target both the treatment of the underlying mental disorder and suicidality [19]. However, there is no consensus on whether antidepressant usage prevents suicide reattempts. Some previous studies suggest that antidepressants may result in an increased risk of suicidality, especially in adolescents [20]. Antipsychotics have been known to have preventive effects on suicidal ideation, attempts, and deaths [21]. However, some longitudinal studies reported antipsychotics has an association with suicide attempts, implying that people taking antipsychotics may have more severe depression which may include psychotic features [7]. Despite the many related studies, the association between reattempts and medication types or the use of polypharmacy remains elusive.

In this study, we aimed to investigate multiple variables including sociodemographic factors and medication usage that influence reattempts among patients who have previously attempted suicide using South Korea's national claim's data. South Korea is known for its high suicide rate, recorded at 24 to 25 deaths per 100 thousand population [22]. Among suicide attempters, patients with depressive disorder were analyzed separately as a subgroup considering their distinct clinical characteristics. By analyzing comprehensive factors affecting suicide reattempts, our objective is to contribute to developing optimal strategies for preventing subsequent suicide attempts.

## Materials and methods

This retrospective cohort study substantially identified the events of study interest (i.e. suicide attempt) using the South Korea's Health Insurance Review and Assessment Service (HIRA) research data derived from claims within the Korean National Health Insurance. The National Health Insurance system in South Korea provides coverage to nearly 97% of the country's population. This extensive coverage enables comprehensive documentation of prescriptions and medical procedures carried out by healthcare institutions for insured individuals, resulting in a comprehensive record of reimbursable medical activities at a national scale. Within psychiatry, most psychotropic prescriptions and consultation fees are typically covered under appropriate diagnoses. For more detailed information about the database, additional references are available for further elucidation [23, 24].

A suicide attempt was defined when an individual presented both suicide-related diagnosis codes and emergency care-related codes, primarily to capture medically severe attempts meeting a minimum threshold of lethality. Suicide-related diagnosis codes not only included R45.8 (other symptoms and signs involving emotional state), X60–X84 (intentional self-harm), Y87 (sequelae of intentional self-harm, and assault and events of undetermined intent), Z64.2 and Z64.3 (problems related to seeking and accepting physical/behavioral, nutritional, and chemical/psychological interventions known to be hazardous and harmful), Z91.5 (personal history of suicide attempt), but also included selected unintentional injuries outlined in S1 Table. This inclusion was based on a comparative analysis of the frequency and methods of suicide attempts with data from the National Emergency Department Information System (NEDIS) [25]. NEDIS independently assesses the intention of injuries, which are not collected in the claim data. Specific unintentional codes were integrated into the cohort entirely, while some were included when accompanied by concurrent psychiatric consultation services during the emergency room visit. The final comparison of the study cohort and 2014 NEDIS suicide attempt data is depicted in S3 Table. The emergency care-related behavior codes have been selected to delineate an individual's visit to the emergency room. A detailed explanaiton of each code can be found in S2 Table. The NN100 code is exclusively applied when a psychiatric consultation service has been conducted within the emergency room setting. To define the subgroup of depressive disorder among total suicide attempters, we used either F32 (depressive episode) or F33 (recurrent depressive disorder) as a primary code.

Patients aged 18 to 100 who had attempted suicide between January 1 and December 31, 2014, were screened for inclusion in this study. To ensure the integrity of our analysis regarding suicide-related codes, specifically concerning unintentional injuries within this demographic, individuals aged over 100 were excluded from the study cohort. The index date was identified as the date when the first suicide attempt was made in 2014, and those with a medical record of three years before the index date and about seven years of follow-up after the index date were included in this study (Fig 1A). As a result, we obtained an anonymized data set of 10 years starting from January 1, 2011, to August 31, 2020, containing variables such as patient demographics, disease diagnoses, comorbidities, types of medications used, and status of healthcare utilization. Medications were grouped as tricyclic antidepressants (TCAs), selective serotonin reuptake inhibitors (SSRIs), serotonin–norepinephrine reuptake inhibitors (SNRIs), noradrenergic and specific serotonergic antidepressants (NaSSAs), monoamine oxidase inhibitors (MAOIs), norepinephrine-dopamine reuptake inhibitors (NDRI), other antidepressants, and atypical antipsychotics.

The evaluation of baseline comorbidities involved calculating the Charlson Comorbidity Index (CCI) score, which was derived from claims records spanning the year preceding the index year. The treatment status was assessed based on the treatment's start date to the end

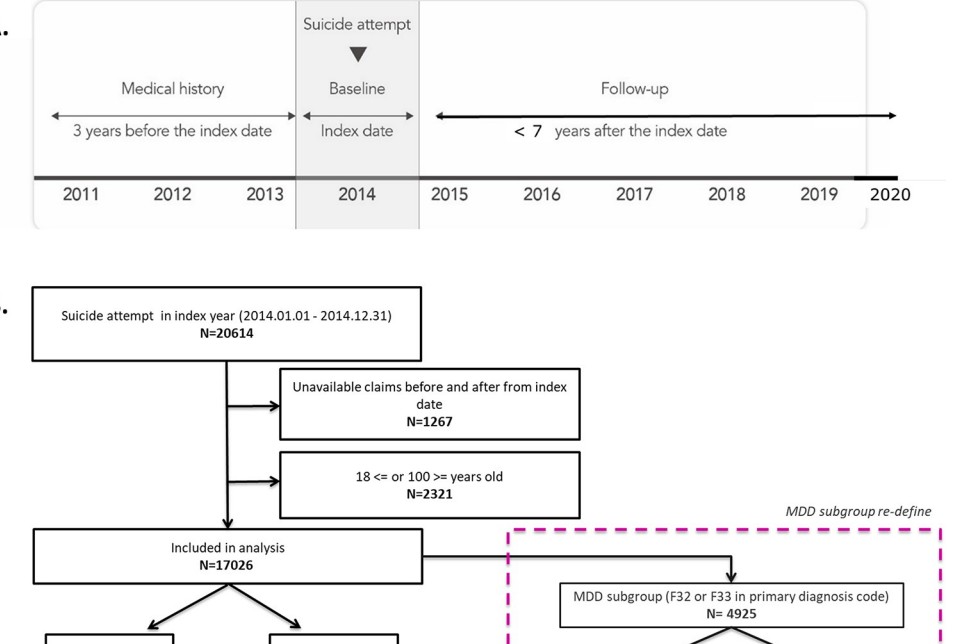

**Fig 1.** The study design (A) and flow of patient selection (B).

date including the medication class, frequency, and duration of treatment for the initial suicide attempt. The frequency of and time to a reattempt after the index date were summarized. For healthcare utilization, the total number of inpatient admissions per person-time, duration of each hospital stay, time to psychiatric inpatient admission, and number of visits to psychiatric or emergency departments after the index date were assessed.

To compare the baseline demographics and clinical characteristics between reattempters and non-reattempters, we used the Mann–Whitney U-test or Student's T-test for continuous variables and the Pearson's Chi-Squared test or Fisher's exact test for categorical variables. We also used the Kaplan-Meier method to estimate the reattempt rate for both overall suicide attempters and those in the depressive disorder subgroup. In addition, a multivariate Cox proportional hazards model was used to assess the risk factors of suicide reattempt. P <0.05 was considered statistically significant. Analyses were performed using the SAS version 9.4 statistical software package (SAS Institute, Cary, NC, USA).

The protocol of the study was exempted from review by Public Institutional Review Board Designated by Ministry of Health and Welfare (IRB number: P01-202010-21-030). All data were fully anonymized before accessed them.

## Results

Of 20,614 identified suicide attempters, 17,026 were included in the analysis after excluding individuals with insufficient claim data before or after the index date, and those outside the specified age range of 18-100(Fig 1B). Overall, 1,853 patients (10.9%) reattempted suicide among the 17,026 suicide attempters within seven years, whereas 15,173 patients did not. Among suicide attempters in the index year, 4,925 patients (28.9%) were diagnosed with

depressive disorder, among whom 936 patients (19.0%) reattempted suicide within seven years (Fig 1B).

The baseline demographics and clinical characteristics of all participants are described in Table 1. The results reveal a higher percentage of female (60.3%) than male (39.7%) suicide attempters; similarly, female patients comprised a higher proportion among reattempters than among non-reattempters (67.2% vs. 59.5%; $p$-value < 0.0001). The mean age of all suicide attempters was 50.1 ±18.27 years, and the mean age of suicide reattempters was younger than that of non-reattempters (45.5 vs. 50.7 years old; $p$-value < 0.0001). Similarly, the proportion of those who lived in metropolitan areas was higher among reattempters than among non-reattempters (48.9% vs. 44.5%; $p$-value = 0.0052). The mean CCI score among suicide attempters was higher than that of non-reattempters (1.7 vs. 1.6, $p$-value = 0.0063).

In addition, the suicide reattempt group had a higher proportion of people who had a history of suicide attempts in the past three years before the index attempt when compared to the suicide non-reattempter group (21.1% vs. 5.7%; $p$-value < 0.0001). The mean number of prior suicide attempts was higher in the suicide reattempters when compared to non-reattempters (1.7 vs. 1.3, $p$-value < 0.0001). Those who had a psychiatric illness (defined by F-code) had a significantly higher percentage of reattempters compared to those without any psychiatric illness (85.9% vs. 65.3%; $p$-value < 0.0001).

**Table 1. Overall baseline and clinical characteristics of suicide attempters.**

| Category | Overall suicide attempters | | | | | |
|---|---|---|---|---|---|---|
| | Total | | Reattempters | | Non-reattempters | P-value |
| | N = 17026 | | N = 1853 | | N = 15173 | |
| **Demographics** | | | | | | |
| Female (%) | 10275 | (60.3) | 1246 | (67.2) | 9029 | (59.5) | <0.0001 |
| Age, years, Mean ± SD | 50.1 | ±18.27 | 45.5 | ±16.80 | 50.7 | ±18.36 | <0.0001 |
| Metropolitan area (Seoul/Gyeonggi; %) | 7659 | (45.0) | 906 | (48.9) | 6753 | (44.5) | 0.0052 |
| **Medical History** | | | | | | |
| CCI Score, Mean ± SD | 1.6 | ±1.98 | 1.7 | ±2.00 | 1.6 | ±1.98 | 0.0063 |
| Having attempted suicide in the past 3 years (%) | 1253 | (7.4) | 391 | (21.1) | 862 | (5.7) | <0.0001 |
| Number of suicide attempts in the past 3 years, Mean ± SD | 1.4 | ±1.04 | 1.7 | ±1.39 | 1.3 | ±0.80 | <0.0001 |
| With psychiatric illness (F-code) (%) | 11495 | (67.5) | 1592 | (85.9) | 9903 | (65.3) | <0.0001 |
| **Psychotropic Medications** | | | | | | |
| Having taken prior medications (%) | 9712 | (57.0) | 1464 | (79.0) | 8248 | (54.4) | <0.0001 |
| Prior medication <2ADs without antipsychotics (%) | 10388 | (61.0) | 671 | (36.2) | 9717 | (64.0) | <0.0001 |
| Prior medication <2ADs with antipsychotics (%) | 1431 | (8.4) | 212 | (11.4) | 1219 | (8.0) | |
| Prior medication > = 2ADs without antipsychotics (%) | 2829 | (16.6) | 411 | (22.2) | 2418 | (15.9) | |
| Prior medication > = 2ADs with antipsychotics (%) | 2378 | (14.0) | 559 | (30.2) | 1819 | (12.0) | |
| More than 3 medications (%) | 5720 | (33.6) | 781 | (42.1) | 4939 | (32.6) | <0.0001 |
| Use of atypical antipsychotics (%) | 6132 | (36.0) | 862 | (46.5) | 5270 | (34.7) | <0.0001 |
| **Medication utilization** | | | | | | |
| No medications (%) | 5756 | (33.8) | 446 | (24.1) | 5310 | (35.0) | <0.0001 |
| Medications prescribed by a non- psychiatrist (%) | 3394 | (19.9) | 316 | (17.0) | 3033 | (20.0) | |
| Medications prescribed by a psychiatrist (%) | 7876 | (46.3) | 1046 | (56.4) | 6830 | (45.0) | |

Note.AD: antidepressants; CCI: Charlson Comorbidity Index; SD: standard deviation

In addition, those who had taken any prior psychiatric medications had a significantly higher percentage among reattempters compared to those among non-reattempters (79.0% vs. 54.4%; *p*-value < 0.0001). The proportion of participants in four subgroups (i.e. less than two antidepressant without antipsychotics, less than two antidepressant with antipsychotics, two or more antidepressants without antipsychotics, and two or more antidepressants with antipsychotics) differed among suicide reattempter group and non-reattempter group (*p*-value < 0.0001). The proportion of patients with more than three medications or the proportion of patients with use of atypical antipsychotics were higher in suicide reattempters when compared to suicide non-reattempters (42.1% vs 32.6%, *p*-value < 0.0001; 46.5% vs. 34.7%, *p*-value < 0.0001). A higher proportion of reattempter group received medications from psychiatrists (56.4% vs 45.0%, p-value < 0.0001).

Demographic and clinical variables among the depressive disorder subgroup are depicted in Table 2. The results showed similar trends of demographic and clinical factors for the overall population. However, a higher percentage of non-reattempters were observed in the depressive disorder subgroup whose psychiatric medications were prescribed by psychiatrists when compared to non-reattempters (69.2% vs. 76.5; *p*-value < 0.0001). Several concurrent medications (more than 3 medications) and the use of antipsychotics did not show a statistically significant difference among reattempters and non-reattempters in this depressive disorder subgroup.

**Table 2. Baseline and clinical characteristics of the depressive disorder subgroup.**

| Category | Suicide attempters with depressive disorder | | | | | | P-value |
| --- | --- | --- | --- | --- | --- | --- | --- |
| | Total | | Reattempters | | Non-reattempters | | |
| | N = 4925 | | N = 936 | | N = 3989 | | |
| **Demographics** | | | | | | | |
| Female (%) | 3525 | (71.6) | 695 | (74.3) | 2830 | (70.9) | 0.0435 |
| Age, years, Mean ± SD | 47.3 | ±17.30 | 43.1 | ±15.43 | 48.3 | ±17.57 | <0.0001 |
| Metropolitan area (Seoul/Gyeonggi; %) | 2231 | (45.3) | 470 | (50.2) | 1761 | (44.1) | 0.1207 |
| **Medical History** | | | | | | | |
| CCI Score, Mean ± SD | 1.7 | ±1.94 | 1.8 | ±1.91 | 1.7 | ±1.95 | 0.3184 |
| Having attempted suicide in the past 3 years (%) | 773 | (15.7) | 284 | (30.3) | 489 | (12.3) | <0.0001 |
| Number of suicide attempts in the past 3 years, Mean ± SD | 1.5 | ±1.11 | 1.7 | ±1.35 | 1.4 | ±0.93 | 0.0009 |
| **Psychotropic medications** | | | | | | | |
| Having taken prior medications (%) | 4791 | (97.3) | 925 | (98.8) | 3866 | (96.9) | 0.0012 |
| Prior medication <2ADs withoutantipsychotics (%) | 971 | (19.7) | 101 | (10.8) | 870 | (21.8) | <0.0001 |
| Prior medication <2ADs with antipsychotics (%) | 404 | (8.2) | 88 | (9.4) | 316 | (7.9) | |
| Prior medication > = 2ADs without antipsychotics (%) | 1758 | (35.7) | 282 | (30.1) | 1476 | (37) | |
| Prior medication > = 2ADs with antipsychotics (%) | 1792 | (36.4) | 465 | (49.7) | 1327 | (33.3) | |
| More than 3 medications (%) | 2805 | (57.0) | 522 | (55.8) | 2283 | (57.2) | 0.4159 |
| Use of atypical antipsychotics (%) | 2839 | (57.6) | 555 | (59.3) | 2284 | (57.3) | 0.2562 |
| **Medication utilization** | | | | | | | |
| No medications (%) | 542 | (11.0) | 110 | (11.8) | 432 | (10.8) | <0.0001 |
| Medications prescribed by a non-psychiatrist (%) | 684 | (13.9) | 178 | (19.0) | 506 | (12.7) | |
| Medications prescribed by a psychiatrist (%) | 3699 | (75.1) | 648 | (69.2) | 3051 | (76.5) | |

Note. Percentages were based on N (the number of patients in the analysis set for each group). AD: antidepressants; CCI: Charlson Comorbidity Index (using one-year data before the index date); SD: standard deviation

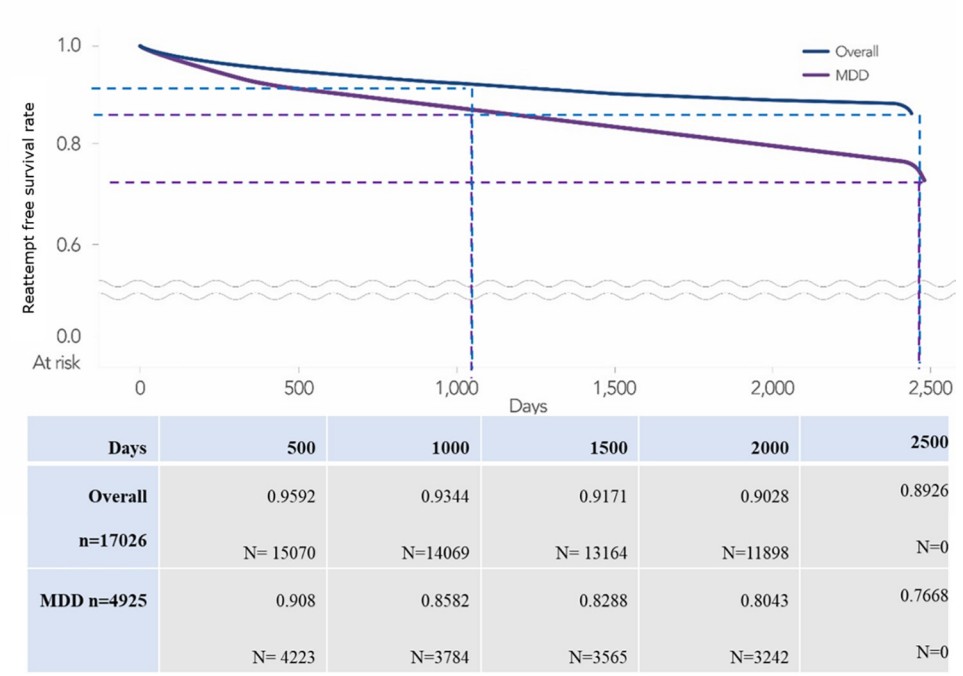

**Fig 2. Kaplan-Meier curve for overall suicide attempters and the depressive disorder subgroup.**

Next, we analyzed the reattempt estimate of suicide attempters using the Kaplan-Meier curve as depicted in Fig 2. The survival probability due to suicide reattempts decreased to 0.9344 after 1000 days from the index attempt and to 0.8926 after 2500 days. However, the survival probability decreased more rapidly to 0.8582 at 1000 days and to 0.7668 at the end of this study in the depressive disorder subgroup. The median time to reattempt was 578 days for overall suicide attempters and 556.6 days for the depressive disorder subgroup.

To further identify individual risk factors of suicide reattempts after the index year, a multivariate Cox proportional hazards model was used for overall suicide attempters described in Table 3 and for the depressive disorder subgroup described in Table 4. The analysis revealed that prior history of suicide attempts, psychiatric illness, diagnosis of depressive disorder, and psychiatric medications increased the risks of suicide reattempts in both overall suicide attempters (hazard ratio [HR] = 2.23, confidence interval [CI] = 1.978,2.517; HR = 1.77, CI = 1.469,2.134; HR = 1.3, CI = 1.154,1.465; HR = 3.2, CI = 2.561,3.996, respectively) and the depressive disorder subgroup (HR = 1.98, CI = 1.716,2.294; HR = 1.77, CI = 1.469, 2.134; HR = 1.30, CI = 1.154,1.465; HR = 3.2, CI = 2.561,3.996, respectively). In particular, the HR significantly rose with medication increase group after the index date when compared to medication decrease group in the overall suicide attempters and depressive disorder subgroup (HR = 19.66, 95% CI: 15.216,25.391; HR = 19.62, 95% CI: 13.472,28.561, respectively). However, when the medications were prescribed by psychiatrists, the HR of suicide reattempts decreased compared to those who did not take any medications among the overall suicide attempters (HR 0.11, 95% CI: 0.093,0.134, $p$-value<0.0001) as well as the depressive disorder subgroup (HR 0.06, 95% CI: 0.047, 0.081, $p$-value<0.0001).Notably, the hazard ratio (HR) for individuals prescribed antidepressants by non-psychiatrists and psychiatrists demonstrated a significant reduction in the risk of suicide reattempt by 78% and 89%, respectively, compared to individuals who did not receive medication (HR 0.22, 95% CI: 0.183, 0.236; HR 0.11, 95% CI: 0.093, 0.134). Within the subgroup with depressive disorder, the reduction in the risk of

**Table 3. Multivariate Cox proportional hazards model for suicide reattempt after the index year among overall suicide attempters.**

| Variable | | Overall suicide attempters | | | |
|---|---|---|---|---|---|
| | | Hazard Ratio | 95% CI | | P-value |
| | | | Lower | Upper | |
| Age (Years) | - | 0.99 | 0.983 | 0.989 | <0.0001 |
| Residential area | Non-metropolitan region | Ref. | - | - | 0.0039 |
| | Metropolitan region | 1.15 | 1.045 | 1.257 | |
| CCI Score | Score:0 | Ref. | - | - | 0.0002 |
| | Score: ≥1 | 1.05 | 1.024 | 1.079 | |
| Suicide attempts in past 3 years | No | Ref. | - | - | <0.0001 |
| | Yes | 2.23 | 1.978 | 2.517 | |
| Diagnosed with psychiatric illness (F-code) | No | Ref. | - | - | <0.0001 |
| | Yes | 1.77 | 1.469 | 2.134 | |
| Diagnosed with depressive disorder | No | Ref. | - | - | <0.0001 |
| | Yes | 1.3 | 1.154 | 1.465 | |
| prior psychotropic medications | No | Ref. | - | - | <0.0001 |
| | Yes | 3.2 | 2.561 | 3.996 | |
| Number of antidepressants with or without antipsychotics | <2ADs without antipsychotics | Ref. | - | - | <0.0001 |
| | Prior medication <2ADs withantipsychotics | 1.97 | 1.632 | 2.378 | |
| | Prior medication > = 2ADs wihtout antipsychotics | 1.61 | 1.338 | 1.931 | |
| | Prior medication > = 2ADs with antipsychotics | 2.42 | 1.868 | 3.142 | |
| Prior psychotropic medications | No | Ref. | - | - | <0.0001 |
| | Yes | 1.16 | 1.08 | 1.253 | |
| Psychotropics taken before and after the index date | Decrease | Ref. | - | - | <0.0001 |
| | No change | 6.85 | 5.443 | 8.616 | |
| | Increase | 19.66 | 15.216 | 25.391 | |
| Prescribed antidepressants | No medications | Ref. | - | - | <0.0001 |
| | By a non-psychiatrist | 0.22 | 0.183 | 0.268 | |
| | By a psychiatrist | 0.11 | 0.093 | 0.134 | |

Note.AD: antidepressants; CCI: Charlson Comorbidity Index

suicide reattempt exhibited a similar trend, as reflected by the hazard ratios (HRs) for prescriptions from both non-psychiatrist and psychiatrist sources at 0.18 and 0.06, respectively when compared with unmedicated group (HR 0.18, 95% CI: 0.132, 0.239; HR 0.06, 95% CI: 0.047, 0.081).)a

## Discussion

In this study, we evaluated the baseline demographics and psychiatric medication usage of suicide attempters using the HIRA database, which covers almost the entire Korean population. A total of 1,853 out of 17,026 (10.9%) suicide attempters were found to have reattempted suicide after the index year. The median time from the index time to the reattempt was 578.0 days, and 21.1% of reattempters demonstrated a history of suicide attempts in the three years before the index date. The multivariate Cox model analysis revealed that prior psychiatric medication and use of multiple psychotropics were highly associated with suicide reattempts in both overall suicide attempters and the depressive disorder subgroup. However, the risk of reattempt was lower if psychiatrists rather than non-psychiatrists prescribed the medications.

There are several important implications from the current findings. First, we found that if there was a suicide attempt before the index attempt, the HR of reattempt after the index

**Table 4. Multivariate Cox proportional hazards model for suicide reattempt after the year among the depressive disorder subgroup.**

| Variable | | Suicide attempters with depressive disorder | | | |
|---|---|---|---|---|---|
| | | Hazard Ratio | 95% CI | | P-value |
| | | | Lower | Upper | |
| Age (Years) | - | 0.98 | 0.979 | 0.988 | <0.0001 |
| Residential area | Non-metropolitan area | Ref. | - | - | 0.0143 |
| | Metropolitan area | 1.18 | 1.033 | 1.339 | |
| CCI Score | Score:0 | Ref. | - | - | 0.0002 |
| | Score: ≥1 | 1.07 | 1.034 | 1.114 | |
| Suicide attempts in past 3 years | No | Ref. | - | - | <0.0001 |
| | Yes | 1.98 | 1.716 | 2.294 | |
| Diagnosed with psychiatric illness (F-code) | No | Ref. | - | - | <0.0001 |
| | Yes | 1.77 | 1.469 | 2.134 | |
| Diagnosed with depressive disorder | No | Ref. | - | - | <0.0001 |
| | Yes | 1.3 | 1.154 | 1.465 | |
| prior psychotropic medications | No | Ref. | - | - | <0.0001 |
| | Yes | 3.2 | 2.561 | 3.996 | |
| Number of antidepressants with or without antipsychotics | <2ADs without antipsychotics | Ref. | - | - | <0.0001 |
| | Prior medication <2ADs withantipsychotics | 5.66 | 2.936 | 10.914 | |
| | Prior medication > = 2ADs withoutantipsychotics | 2.67 | 1.974 | 3.609 | |
| | Prior medication > = 2ADs with antipsychotics | 2 | 1.566 | 2.562 | |
| Prior psychotropic medications | No | Ref. | - | - | <0.0001 |
| | Yes | 3.18 | 2.472 | 4.084 | |
| Psychotropics taken before and after the index date | Decrease | Ref. | - | - | <0.0001 |
| | No change | 15.28 | 10.767 | 21.688 | |
| | Increase | 19.62 | 13.472 | 28.561 | |
| Prescribed antidepressants | No medications | Ref. | - | - | <0.0001 |
| | By a non-psychiatrist | 0.18 | 0.132 | 0.239 | |
| | By a psychiatrist | 0.06 | 0.047 | 0.081 | |

Note.AD: antidepressants; CCI: Charlson Comorbidity Index

attempt becomes 2.23 times greater when compared to those without previous suicide attempts before the index event (95% CI: 1.978, 2.517, p-value <0.0001). This suggests that the people who make multiple suicide attempts should be closely observed due to their high risk of reattempts. Our analysis suggests that the optimal duration of close observation should last between 6 and 24 months for those with previous suicide attempts considering the median time to reattempt was shorter (556.5 days) in the depressive disorder subgroup compared to the overall group (578 days).

Second, 19% of those in the depressive disorder subgroup reattempted suicide, which was nearly double the reattempts among all suicide attempters (10.9%). This suggests that depressive disorder is a major risk factor for suicide as described in previous studies [15, 26, 27]. Notably, within the depressive disorder subgroup, a significant proportion of participants were prescribed two or more antidepressants years before the index date, and their medical records show concurrent usage of three or more antidepressants. This implies that their disease status was severe, and the possibility of treatment-resistant depression (TRD) emerged considering its definition as an inadequate response even after more than two qualifying antidepressants with or without antipsychotic use for an adequate treatment duration [28]. Several lines of research have shown that TRD patients not only suffer from higher medical costs than

patients with non-TRD but also have a higher risk of suicide attempts and completed suicide rates [29–31]. These results point to the importance of screening the severity of depression as early as possible along with treatment resistance to reduce suicidal risk for depressive disorder patients in routine clinical practice.

Third, the subgroups with a history of multiple psychotropics use had a higher risk of suicide reattempts. Patients who were prescribed two or more antidepressants with antipsychotics had a 3.18 times higher risk of suicide reattempts (95% CI: 2.472, 4.084, $p$-value $<0.0001$) than those who had been treated with less than two types of antidepressants. This indicates that the risk of suicide does not reduce even after patients are treated with medications and multiple psychotropics, possibly due to the residual symptoms from their primary diagnosis -depressive disorder. Thus, clinical interventions for suicidality should be more frequent in patients who use multiple types of drugs, and their symptoms should be effectively addressed.

Fourth, our study suggests that when psychiatric medications were prescribed by psychiatrists, the risk of suicide reattempt was lower than when psychiatric medications were prescribed by non-psychiatrists especially in the depressive disorder subgroup. A few previous studies also reveal that antidepressants prescribed by different medical providers lead to different outcomes and adherence patterns in treating depressive disorder [32–34]. The current study suggests that prescriptions by psychiatrists are associated with lower risks of reattempts in overall attempters (HR = 0.11, 95% CI = 0.093,0.134), and even lower risks in suicidal patients with depressive disorder (HR = 0.06, 95% CI = 0.017,0.081) when compared with people who did not take medications after suicide attempt. Therefore, it would be advisable to refer individuals suffering from severe or acute suicidality to psychiatrists for evaluation and treatment optimization.

## Limitation

The present study has several limitations. As it relied on claim-based data, the study population was selected solely based on diagnostic codes, and not on their clinical evaluation. Although efforts were made to validate the diagnosis of suicide attempts by cross-referencing additional independent national data sources, it is essential to acknowledge that the categorization of the suicide attempt group remains operational, potentially resulting in some degree of misclassification. In addition, we were unable to assess adherence to antidepressant medications as claim data cannot capture the extent to which patients follow prescribed medication regimens. The crucial doctor-patient relationship, which plays a significant role in medication adherence, also could not be assessed within the scope of this investigation.

## Conclusion

The burden of suicide remains underestimated. The current study comprehensively investigated the risk factors of suicidal reattempts from the perspective of clinical, pharmacological, and in-service providers'. Our study results revealed various risk factors associated with suicide reattempts among suicide attempters and suggested that patients with depressive disorder should be monitored closely. Future research should characterize risk groups based on their treatment history to assess TRD and develop individualized prevention strategies for potential harms or risks of suicide attempts.

## Supporting information

**S1 Checklist. Human participants research checklist.**
(DOCX)

**S1 Table. Suicide-related diagnoses and categorization.**
(DOCX)

**S2 Table. Emergency care-related codes.**
(DOCX)

**S3 Table. Comparison of the cohort with 2014 NEDIS data.**
(DOCX)

## Author Contributions

**Conceptualization:** Min Ji Kim, Min Jung Koh, Yong Min Ahn.

**Data curation:** Jeong Hun Yang, Min Jung Koh, Youngdoe Kim.

**Formal analysis:** Min Ji Kim, Jeong Hun Yang, Youngdoe Kim.

**Investigation:** Youngdoe Kim, Bolam Lee.

**Methodology:** Youngdoe Kim, Bolam Lee, Yong Min Ahn.

**Project administration:** Jeong Hun Yang, Bolam Lee.

**Supervision:** Min Ji Kim, Yong Min Ahn.

**Visualization:** Min Ji Kim, Bolam Lee.

**Writing – original draft:** Min Ji Kim, Bolam Lee.

**Writing – review & editing:** Min Ji Kim, Jeong Hun Yang, Min Jung Koh, Bolam Lee, Yong Min Ahn.

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
