## [Decision Letter · Decision Letter 0]

27 Nov 2023

PONE-D-23-32199Risk factors of reattempt among suicide attempters in South Korea: A nationwide retrospective cohort studyPLOS ONE

Dear Dr. Ahn,

Thank you for submitting your manuscript to PLOS ONE. After careful consideration, we feel that it has merit but does not fully meet PLOS ONE’s publication criteria as it currently stands. Therefore, we invite you to submit a revised version of the manuscript that addresses the points raised during the review process.

As you will see, both reviewers found your work important providing novel insights yet they recommend a number of amendments to consider and provide helpful suggestions. Please respond to all of them. 

We look forward to receiving your revised manuscript.

Kind regards,

Xenia Gonda

Academic Editor

PLOS ONE

 [This study was funded by Janssen Korea Ltd].  

[I have read the journal's policy and the authors of this manuscript have the following competing interests: MJ Koh, Y Kim, and B Lee are employees of Janssen Korea Ltd. YM Ahn declared a research fund from Janssen Korea Ltd. and participated in speakers’ events in Janssen Korea Ltd, Lundbeck Korea Co., Ltd., and Korea Otsuka Pharmaceutical. The remaining authors have nothing to disclose.]. 

Additional Editor Comments:

As you will see, both reviewers found your work important providing novel insights yet they recommend a number of amendments to consider and provide helpful suggestions. Please respond to all of them.

Reviewers' comments:

Reviewer's Responses to Questions

**Comments to the Author**

1. Is the manuscript technically sound, and do the data support the conclusions?

Reviewer #1: No

Reviewer #2: Partly

2. Has the statistical analysis been performed appropriately and rigorously? 

Reviewer #1: Yes

Reviewer #2: Yes

3. Have the authors made all data underlying the findings in their manuscript fully available?

Reviewer #1: No

Reviewer #2: No

4. Is the manuscript presented in an intelligible fashion and written in standard English?

Reviewer #1: Yes

Reviewer #2: Yes

5. Review Comments to the Author

Reviewer #1: The goal of this study was to use a claims database in Korea to identify the demographic and clinical characteristics of individuals with a history of suicide attempts. By examining medical records for the 3 years before the index date and tracking data for 7 years afterward, it was found that 10% of the 17,026 suicide attempters made another attempt, and 29% were diagnosed with Major Depressive Disorder (MDD). Notably, the study suggests that the risk of a repeated attempt is lower when the treatment is prescribed by a psychiatrist in MDD patients.

I believe that this paper, which has implications for policy, is timely and should be published promptly.

However, one noteworthy aspect is that, as this paper is not exclusively intended for Korean readers, an additional explanation is essential in the introduction. This explanation should clarify why conducting the study in Korea is significant globally. To enhance its appeal, the paper could include information on Korea's higher suicide rate compared to other countries.

Thank you.

Reviewer #2: This is a study examining risks factors related to reattempt of suicide in a cohort of roughly 17,000 patients in Korea with a ‘suicide attempt’ recorded from health insurance claims, in 2014, as well as separately in a sub group of these patients with a history of depression. Retrospective data analysis has then followed them for a period of 6-7 years following the 2014 suicide attempt, to see if they reattempted, and examined risks related to demographic factors, medical history, psychotropic medication and timing.

This work does appear to add helpful insights to the existing literature. On the whole, the conclusions drawn do correlate with the data presented, although I have concerns that some of the hazard ratios given need better placing in the context of the relevant reference groups used, which has not always been done.

My main concern here relates to the diagnosis codes used. It may be that with additional explanation this is ok, but based on the current explanation given, I feel these cases likely overestimate actual suicide attempts. I also have a concern about the naming/diagnosis of the ‘major’ depression group. See below for more detailed comments.

Abstract

1. “The risk of reattempt was lower if psychiatrists rather than non-psychiatrists prescribed medications (HR 0.11, 95% CI: 0.093, 0.134, p-value <0.01).” I find this statement misleading as it suggests the risk is 89% lower when psychiatrists prescribed medication, compared to non-psychiatrists. This is not the case, as patients not prescribed medication are the reference group here. This needs rewording.

Introduction – this appears thorough and well written with a reasonable explanation of the existing evidence and context into which this work has been carried out.

Methods

2. The wording around how the database is outlined (page 5, paragraph 1) could do with updating to make clear exactly what this database is. I presume this is for claims to the National Health Insurance scheme, but it would be helpful to outline this.

3. For those not familiar with how the Korean national health insurance system works, I think it needs a brief outline of what health data/healthcare contacts this data will or won’t record. Is it any and all healthcare contacts, or are there some instances that would not be captured in this data?

4. By suicide attempt definition needing to include BOTH a diagnosis code and emergency care code, might this exclude suicide attempts that require a lower level of healthcare than “emergency care”? Again this needs outlining, or at least acknowledging in the limitations if this indeed is a danger.

5. I have significant concerns about the diagnosis codes used. This may be explained by having further detail of what the ‘emergency care codes’ mean, but at present a large number of the ICD-10 codes outlined, to my mind do not by themselves indicate a suicide attempt, as there is no intent attached to them. While including some with ‘undetermined’ intent (such as Y10-Y34) would be justified (with referencing), including a lot with no intent recorded at all, potentially means that a large number of non-suicide instances of harm may have been captured here. This includes ‘unspecified falls’, all the S codes included, T14.9 (unspecified injury), F11-F19, T30-T65. Depending on what the ‘emergency care codes’ refer to (see point 5), this may provide additional detail to indicate these injuries to be suicide attempts, but this definitely needs further explanation/justification.

6. To the point above, there is no mention of what the emergency care related codes mean – this definitely needs including (it may be a brief explanation in the text and more detailed information in a supplementary table). At present, the reader has no way of contextualising what these codes mean, which does not help with establishing whether the codes used do likely record actual suicide attempts or not.

7. I also have a concern that the codes used for MDD are actually capturing ANY depressive illness (including mild-moderate ones), rather than only MAJOR depressive illness. This either needs further explanation (for instance is it only F32.2/F32.3 and F33.2/F33.3?), or if all categories of depression are included, potentially renaming this category to ‘depressive disorder’, rather than MDD.

8. Why have patients >100 years old been excluded? This is likely to be very small numbers but again I feel needs explaining/justifying.

9. At the top of page 6 “the first suicide attempt was committed…”. I do not feel it is appropriate to use the word “committed” here, which implies illegality. It should be referred to as “made”.

10. Better to use “anonymised” datatset, rather than “de-identified”

11. I am confused about the statements relating to the ethical approval for this study and this certainly needs clarification. On page 5 it states, “The protocol of the study was exempted form review by Public Institutional Review Board Designated by Ministry of Health and Welfare”, but page 6 states “The institutional review board reviewed and approved the study protocol before the study was conducted.” Which of these statements is correct, as it feels that they cannot both be?

12. “The Charlson Comorbidity Index (CCI) score was used to assess baseline comorbidities using medical records from the past year of the index year.” I am a bit confused by this sentence. Does it mean to look for baseline comorbidities recorded in the year before the index year? Or perhaps something else?

Results

13. In the first line referring to numbers included from those screened, reference should be made to figure 1B to see an explanation of those excluded.

14. The footnotes to table 1 should include explanation of the terms “w/” and “w/o” (presumably with and without), or just include the full words in the table.

15. I am confused about the first grouping for psychotropic medication use. In table 1 this group is called “<2” antidepressants with (or without) antipsychotics. However in the text, it refers to “one” antidepressant with (or without) antipsychotics. “<2” would imply it also includes individuals taking no medication, as well as those taking one antidepressant, however the text suggests it is only those taking one. This needs clarifying and consistent naming within the table and text.

16. In table 3 & 4, the category “having taken any prior medications”, is presumably any prior psychotropic medications. I think this needs including in the table if it is.

17. On page 11, “In particular, the HR significantly rose with medications after the index date in the overall suicide attempters and MDD subgroup (HR=19.66, 95% CI: 15.216,25.391; HR= 19.62, 95% CI: 13.472,28.561, respectively)” seems a little misleading. What you are comparing in this hazard ratio is those who had no change in their previous psychotropic medication, or an increase in their previous psychotropic medication, compared to those who had a decrease in their psychotropic medication (the reference group) after the index date. The sentence above needs to include “compared to…” and more accurately outline what you are actually comparing here.

18. On page 11, the following wording is I think unhelpful: “More interestingly, the HR of the prescriptions for antidepressants by psychiatrists was two to three times lower than the HR of prescriptions by non-psychiatrists (HR 0.22, 95% CI: 0.183, 0.268, p-value <0.0001) as well as the MDD subgroup (HR 0.18, 95% CI: 0.132, 0.239, p-value<0.0001).” Only the HR of the prescriptions by non-psychiatrists is given in both cases, which makes it sound as though this is the HR for antidepressants by psychiatrists. I think both HR (given by psychiatrists or non-psychiatrists) should be given to aid the reader understanding what the difference in HRs is here. This also needs to state that it is “when compared to individuals prescribed no psychotropic medication”, as again what your reference group is here is very important.

Discussion

19. The discussion of lower risk for those prescribed antidepressants by a psychiatrist (at the bottom of page 15/top of page 16) again I feels needs further context adding as to the comparison group here, which is individuals prescribed NO antidepressant, not individuals prescribed an antidepressant by a non-psychiatrist.

20. The limitations section certainly needs to add discussion around the codes used and risks of misclassification, as per comments 4-7 on the methods.

6. PLOS authors have the option to publish the peer review history of their article (what does this mean?). If published, this will include your full peer review and any attached files.

Reviewer #1: No

Reviewer #2: No

---

## [Author Response · Author response to Decision Letter 0]

16 Jan 2024

<Response to Editor comments >

We revised and added statements based on the journal requirements:

1. We made revision to fit the PLOS ONE style requirement. 

2. For ethics statement, we revised as the following: 

The protocol of the study was exempted from review by Public Institutional Review Board Designated by Ministry of Health and Welfare (IRB number: P01-202010-21-030). All data were fully anonymized before accessed them.

3. For financial disclosure, we stated the role of the funder as well: 

This study was funded by Janssen Korea Ltd. The funder had no role in study design, data collection and analysis, decision to publish, or preparation of the manuscript.

4. For the competing interest section, we added that we adhere to PLOS ONE policies.

MJ Koh, Y Kim, and B Lee are employees of Janssen Korea Ltd. YM Ahn declared a research fund from Janssen Korea Ltd. and participated in speakers’ events in Janssen Korea Ltd, Lundbeck Korea Co., Ltd., and Korea Otsuka Pharmaceutical. The remaining authors have nothing to disclose. This does not alter our adherence to PLOS ONE policies on sharing data and materials.

5. In, Data Availability statement, we added more information to apply and gain the data from HIRA database: 

Data can be obtained via website of HIRA(HIRA bigdata open portal) by filling out the application. The data is provided in a DVD (text file) format and a fee for the data is subject to be charged. More detailed information on how to access the database is in the following website: 

https://opendata.hira.or.kr/op/opc/selectOpenDataAplInfoView.do

<Response to reveiwers>

We express our deepest gratitude and appreciation to the editor and the reviewers for providing their valuable insights and helpful comments. We have carefully considered their comments and substantially revised our paper. Detailed responses to the reviewers’ comments are provided below. Any revisions and additions to our manuscript are indicated in red font.

Reviewer #1: The goal of this study was to use a claims database in Korea to identify the demographic and clinical characteristics of individuals with a history of suicide attempts. By examining medical records for the 3 years before the index date and tracking data for 7 years afterward, it was found that 10% of the 17,026 suicide attempters made another attempt, and 29% were diagnosed with Major Depressive Disorder (MDD). Notably, the study suggests that the risk of a repeated attempt is lower when the treatment is prescribed by a psychiatrist in MDD patients.

I believe that this paper, which has implications for policy, is timely and should be published promptly.

However, one noteworthy aspect is that, as this paper is not exclusively intended for Korean readers, an additional explanation is essential in the introduction. This explanation should clarify why conducting the study in Korea is significant globally. To enhance its appeal, the paper could include information on Korea's higher suicide rate compared to other countries.

Answer> Thank you for your comment. We added the following to the Introduction to include information on Korea's higher suicide rate. 

In this study, we aimed to investigate multiple variables including sociodemographic factors and medication usage that influence reattempts among patients who have previously attempted suicide using South Korea’s national claim’s data. South Korea is known for its high suicide rate, recorded at 24 to 25 deaths per 100 thousand population [22]. Among suicide attempters, patients with depressive disorder were analyzed separately as a subgroup considering their distinct clinical characteristics. By analyzing comprehensive factors affecting suicide reattempts, our objective is to can contribute to developing optimal strategies for preventing subsequent suicide attempts.

Reviewer #2: This is a study examining risks factors related to reattempt of suicide in a cohort of roughly 17,000 patients in Korea with a ‘suicide attempt’ recorded from health insurance claims, in 2014, as well as separately in a sub group of these patients with a history of depression. Retrospective data analysis has then followed them for a period of 6-7 years following the 2014 suicide attempt, to see if they reattempted, and examined risks related to demographic factors, medical history, psychotropic medication and timing.

 This work does appear to add helpful insights to the existing literature. On the whole, the conclusions drawn do correlate with the data presented, although I have concerns that some of the hazard ratios given need better placing in the context of the relevant reference groups used, which has not always been done.

 My main concern here relates to the diagnosis codes used. It may be that with additional explanation this is ok, but based on the current explanation given, I feel these cases likely overestimate actual suicide attempts. I also have a concern about the naming/diagnosis of the ‘major’ depression group. See below for more detailed comments.

Abstract

1. “The risk of reattempt was lower if psychiatrists rather than non-psychiatrists prescribed medications (HR 0.11, 95% CI: 0.093, 0.134, p-value <0.01).” I find this statement misleading as it suggests the risk is 89% lower when psychiatrists prescribed medication, compared to non-psychiatrists. This is not the case, as patients not prescribed medication are the reference group here. This needs rewording.

Answer> Thank you for your comment. We revised the sentence as following.

 The risk of reattempt decreased in individuals receiving antidepressant prescriptions compared to those unmedicated, showing a reduction of 78% when prescribed by non-psychiatrists and 89% when prescribed by psychiatrists.

Methods

2. The wording around how the database is outlined (page 5, paragraph 1) could do with updating to make clear exactly what this database is. I presume this is for claims to the National Health Insurance scheme, but it would be helpful to outline this.

Answer> Thank you for the comment. I revised the paragraph considering the comment #3: 

This retrospective cohort study substantially identified the events of study interest (i.e. suicide attempt) using the South Korea’s Health Insurance Review and Assessment Service (HIRA) research data derived from claims within the Korean National Health Insurance. The National Health Insurance system in South Korea provides coverage to nearly 97% of the country's population. This extensive coverage enables comprehensive documentation of prescriptions and medical procedures carried out by healthcare institutions for insured individuals, resulting in a comprehensive record of reimbursable medical activities at a national scale. Within psychiatry, most psychotropic prescriptions and consultation fees are typically covered under appropriate diagnoses. For more detailed information about the database, additional references are available for further elucidation [23,24].

3. For those not familiar with how the Korean national health insurance system works, I think it needs a brief outline of what health data/healthcare contacts this data will or won’t record. Is it any and all healthcare contacts, or are there some instances that would not be captured in this data?

Answer> We added some details on Korean healthcare system and what it records in the field of psychiatry. In psychiatry, most psychotropic prescriptions and consultation fees are typically covered under suitable diagnoses. Certain patients may choose to forgo insurance coverage to prevent the recording of their prescription data due to societal stigma. However, opting out of insurance coverage significantly increases healthcare costs, rendering it less feasible for a substantial number of individuals to consider this alternative. The precise extent of these increased expenses remains uncertain and cannot be reliably anticipated due to the lack of recorded information on such cases. The revised paragraph is same as comment #2.

This retrospective cohort study substantially identified the events of study interest (i.e. suicide attempt) using the South Korea’s Health Insurance Review and Assessment Service (HIRA) research data derived from claims within the Korean National Health Insurance. The National Health Insurance system in South Korea provides coverage to nearly 97% of the country's population. This extensive coverage enables comprehensive documentation of prescriptions and medical procedures carried out by healthcare institutions for insured individuals, resulting in a comprehensive record of reimbursable medical activities at a national scale. Within psychiatry, most psychotropic prescriptions and consultation fees are typically covered under appropriate diagnoses. For more detailed information about the database, additional references are available for further elucidation [23,24].

4. By suicide attempt definition needing to include BOTH a diagnosis code and emergency care code, might this exclude suicide attempts that require a lower level of healthcare than “emergency care”? Again this needs outlining, or at least acknowledging in the limitations if this indeed is a danger.

Answer> Thank you for the comment. We included emergency care code to set the severity threshold for medically serious suicide attempt. Intention and lethality of suicide are both important aspects of suicide attempt[1]. While some studies have indicated an association between these factors[2, 3], others argues that they are two distinct construct[4]. With only claim data we are not able to assess the intention. Only the construct of lethality was included to define the cohort. I revised as follows;

A suicide attempt was defined when an individual presented both suicide-related diagnosis codes and emergency care-related codes, primarily to capture medically severe attempts meeting a minimum threshold of lethality. 

5. I have significant concerns about the diagnosis codes used. This may be explained by having further detail of what the ‘emergency care codes’ mean, but at present a large number of the ICD-10 codes outlined, to my mind do not by themselves indicate a suicide attempt, as there is no intent attached to them. While including some with ‘undetermined’ intent (such as Y10-Y34) would be justified (with referencing), including a lot with no intent recorded at all, potentially means that a large number of non-suicide instances of harm may have been captured here. This includes ‘unspecified falls’, all the S codes included, T14.9 (unspecified injury), F11-F19, T30-T65. Depending on what the ‘emergency care codes’ refer to (see point 5), this may provide additional detail to indicate these injuries to be suicide attempts, but this definitely needs further explanation/justification.

Answer> Appreciating your insights on diagnostic codes, it's evident that our study's description of the selection process for diagnostic codes defining the study population was insufficient. Solely employing intentional codes like X codes resulted in a significant underestimation of suicide attempts compared to data from the NEDIS (National Emergency Department Information System) database in South Korea. NEDIS, established in 2003, collects real-time medical information independently from claim data[5]. In order to align our cohort with the 2014 NEDIS database, we meticulously assembled all unintentional injuries and handpicked specific categories to ensure their numbers and proportions corresponded to those recorded in the NEDIS dataset. For instance, the aggregate of all intentional and unintentional hangings closely resembled the reported number of hanging suicide attempts in NEDIS 2014. However, in cases like drug intoxication, unintentional codes outnumbered NEDIS's suicide attempts’ drug intoxication category. Thus, we included only those with drug intoxication who received psychiatric service consultation during their emergency room stay (indicated by simultaneous injury and NN100 code). We further detailed the rationale behind including undetermined intent codes and provided comparative results with NEDIS for comprehensive clarity in supplementary tables 1~3.

Suicide-related diagnosis codes not only included R45.8 (other symptoms and signs involving emotional state), X60–X84 (intentional self-harm), Y87 (sequelae of intentional self-harm, and assault and events of undetermined intent), Z64.2 and Z64.3 (problems related to seeking and accepting physical/behavioral, nutritional, and chemical/psychological interventions known to be hazardous and harmful), Z91.5 (personal history of suicide attempt), but also included selected unintentional injuries outlined in Supplementary Table 1. This inclusion was based on a comparative analysis of the frequency and methods of suicide attempts with data from the 2014 National Emergency Department Information System (NEDIS). NEDIS independently assesses the intention of injuries, which are not collected in the claim data. Specific unintentional codes were integrated into the cohort entirely, while some were included when accompanied by concurrent psychiatric consultation services during the emergency room visit. The final comparison of the study cohort and 2014 NEDIS suicide attempt data is depicted in Supplementary table 3.

6. To the point above, there is no mention of what the emergency care related codes mean – this definitely needs including (it may be a brief explanation in the text and more detailed information in a supplementary table). At present, the reader has no way of contextualising what these codes mean, which does not help with establishing whether the codes used do likely record actual suicide attempts or not.

Answer> Under South Korea's fee-for-service framework, an administrative charge is imposed on all emergency room visits. This fee acts as a validation point for confirming the incidence of an emergency room visit. I removed the list of codes on manuscript and added further explanation of the each emergency code related to the South Korea’s healthcare system on supplementary table2. 

The emergency care-related behavior codes have been selected to delineate an individual's visit to the emergency room. A detailed explanation of each code can be found in Supplementary Table 2. The NN100 code is exclusively applied when a psychiatric consultation service has been conducted within the emergency room setting. 

7. I also have a concern that the codes used for MDD are actually capturing ANY depressive illness (including mild-moderate ones), rather than only MAJOR depressive illness. This either needs further explanation (for instance is it only F32.2/F32.3 and F33.2/F33.3?), or if all categories of depression are included, potentially renaming this category to ‘depressive disorder’, rather than MDD.

Answer> I agree to your feedback and have made the revision throughout the whole manuscript, consistently changing the “MDD subgroup” as the "depressive disorder subgroup."

8. Why have patients >100 years old been excluded? This is likely to be very small numbers but again I feel needs explaining/justifying.

Answer> We opted to exclude individuals aged over 100 from our analysis due to the increased complexity arising from undetermined intent codes within this demographic even though the numerical representation within this age bracket was deemed to be negligible in our study.

Patients aged 18 to 100 who had attempted suicide between January 1 and December 31, 2014, were screened for inclusion in this study. To ensure the integrity of our analysis regarding suicide-related codes, specifically concerning unintentional injuries within this demographic, individuals aged over 100 were excluded from the study cohort.

9. At the top of page 6 “the first suicide attempt was committed…”. I do not feel it is appropriate to use the word “committed” here, which implies illegality. It should be referred to as “made”.

Answer>Thank you for the comment. I changed “commited” as “made”

The index date was identified as the date when the first suicide attempt was made in 2014,

10. Better to use “anonymised” datatset, rather than “de-identified”

Answer>Thank you for the comment. I changed “de-identifeid” as “anonymized”

As a result, we obtained an anonymized data set of 10 years starting from January 1, 2011, to August 31, 2020.

11. I am confused about the statements relating to the ethical approval for this study and this certainly needs clarification. On page 5 it states, “The protocol of the study was exempted form review by Public Institutional Review Board Designated by Ministry of Health and Welfare”, but page 6 states “The institutional review board reviewed and approved the study protocol before the study was conducted.” Which of these statements is correct, as it feels that they cannot both be?

Answer>Thank you for the comment. The first one is the right one and I erased the latter. This study was exempted from review by IRB since it is an anonymized public database. The sentence has been relocated to the end of the Method section for readability.

The protocol of the study was exempted form review by Public Institutional Review Board Designated by Ministry of Health and Welfare (IRB number: P01-202010-21-030). All data were fully anonymized before accessed them.

12. “The Charlson Comorbidity Index (CCI) score was used to assess baseline comorbidities using medical records from the past year of the index year.” I am a bit confused by this sentence. Does it mean to look for baseline comorbidities recorded in the year before the index year? Or perhaps something else?

Answer> You have understood correctly. We have revised the manuscript to enhance clarity and convey the intended meaning more effectively.

The evaluation of baseline comorbidities involved calculating the Charlson Comorbidity Index (CCI) score, which was derived from claims records spanning the year preceding the index year.

Results

13. In the first line referring to numbers included from those screened, reference should be made to figure 1B to see an explanation of those excluded.

Answer> Thank you for the comment. We revised the manuscript to include how people were excluded from the study population. 

Of 20,614 identified suicide attempters, 17,026 were included in the analysis after excluding individuals with insufficient claim data before or after the index date, and those outside the specified age range of 18-100.(Fig 1B).

14. The footnotes to table 1 should include explanation of the terms “w/” and “w/o” (presumably with and without), or just include the full words in the table.

Answer> Thank you for the comment. We revised all tables contain “w/” and “w/o” to full words. 

Psychotropic Medications

Having taken prior medications (%) 9712 (57.0) 1464 (79.0) 8248 (54.4) <0.0001

Prior medication <2ADs without antipsychotics (%) 10388 (61.0) 671 (36.2) 9717 (64.0) <0.0001

Prior medication <2ADs with antipsychotics (%) 1431 (8.4) 212 (11.4) 1219 (8.0) 

Prior medication >=2ADs without antipsychotics (%) 2829 (16.6) 411 (22.2) 2418 (15.9) 

Prior medication >=2ADs with antipsychotics (%) 2378 (14.0) 559 (30.2) 1819 (12.0) 

More than 3 medications (%) 5720 (33.6) 781 (42.1) 4939 (32.6) <0.0001

Use of atypical antipsychotics (%) 6132 (36.0) 862 (46.5) 5270 (34.7) <0.0001

15. I am confused about the first grouping for psychotropic medication use. In table 1 this group is called “<2” antidepressants with (or without) antipsychotics. However in the text, it refers to “one” antidepressant with (or without) antipsychotics. “<2” would imply it also includes individuals taking no medication, as well as those taking one antidepressant, however the text suggests it is only those taking one. This needs clarifying and consistent naming within the table and text.

Answer> Sorry for the confusion. I revised the manuscript as following;

(i.e. less than two antidepressant without antipsychotics, less than two antidepressant with antipsychotics, two or more antidepressants without antipsychotics, and two or more antidepressants with antipsychotics)

16. In table 3 & 4, the category “having taken any prior medications”, is presumably any prior psychotropic medications. I think this needs including in the table if it is.

Answer> Thank you for your feedback. I have revised table 3&4 to ensure clarity in their meaning.

prior psychotropic medications No Ref. - - <0.0001

 Yes 3.2 2.561 3.996 

17. On page 11, “In particular, the HR significantly rose with medications after the index date in the overall suicide attempters and MDD subgroup (HR=19.66, 95% CI: 15.216,25.391; HR= 19.62, 95% CI: 13.472,28.561, respectively)” seems a little misleading. What you are comparing in this hazard ratio is those who had no change in their previous psychotropic medication, or an increase in their previous psychotropic medication, compared to those who had a decrease in their psychotropic medication (the reference group) after the index date. The sentence above needs to include “compared to…” and more accurately outline what you are actually comparing here.

Answer> Thank you for the comment. I revised the part to clarify the comparison group. 

In particular, the HR significantly rose with medication increase group after the index date when compared to medication decrease group in both the overall suicide attempters and depressive disorder subgroup (HR=19.66, 95% CI: 15.216,25.391; HR= 19.62, 95% CI: 13.472,28.561, respectively).

18. On page 11, the following wording is I think unhelpful: “More interestingly, the HR of the prescriptions for antidepressants by psychiatrists was two to three times lower than the HR of prescriptions by non-psychiatrists (HR 0.22, 95% CI: 0.183, 0.268, p-value <0.0001) as well as the MDD subgroup (HR 0.18, 95% CI: 0.132, 0.239, p-value<0.0001).” Only the HR of the prescriptions by non-psychiatrists is given in both cases, which makes it sound as though this is the HR for antidepressants by psychiatrists. I think both HR (given by psychiatrists or non-psychiatrists) should be given to aid the reader understanding what the difference in HRs is here. This also needs to state that it is “when compared to individuals prescribed no psychotropic medication”, as again what your reference group is here is very important.

Answer> Thank you for the comment. I revised the part to clarify the comparison group. 

Notably, the hazard ratio (HR) for individuals prescribed antidepressants by non-psychiatrists and psychiatrists demonstrated a significant reduction in the risk of suicide reattempt by 78% and 89%, respectively, compared to individuals who did not receive medication (HR 0.22, 95% CI: 0.183, 0.236; HR 0.11, 95% CI: 0.093, 0.134). Within the subgroup with depressive disorder, the reduction in the risk of suicide reattempt exhibited a similar trend, as reflected by the hazard ratios (HRs) for prescriptions from both non-psychiatrist and psychiatrist sources at 0.18 and 0.06, respectively when compared with unmedicated group (HR 0.18, 95% CI: 0.132, 0.239; HR 0.06, 95% CI: 0.047, 0.081).

Discussion

19. The discussion of lower risk for those prescribed antidepressants by a psychiatrist (at the bottom of page 15/top of page 16) again I feels needs further context adding as to the comparison group here, which is individuals prescribed NO antidepressant, not individuals prescribed an antidepressant by a non-psychiatrist.

Answer> Thank you for the comment. I added the comparison group to clarify the meaning.

The current study suggests that prescriptions by psychiatrists are associated with lower risks of reattempts in overall attempters (HR=0.11, 95% CI=0.093,0.134), and even lower risks in suicidal patients with depressive disorder (HR=0.06, 95% CI=0.017,0.081) when compared with people who did not take medications after suicide attempt.

20. The limitations section certainly needs to add discussion around the codes used and risks of misclassification, as per comments 4-7 on the methods.

Answer> Thank you for the comment. I added those in the limitation part. 

As it relied on claim-based data, the study population was selected solely based on diagnostic codes, and not on their clinical evaluation. Although efforts were made to validate the diagnosis of suicide attempts by cross-referencing additional independent national data sources, it is essential to acknowledge that the categorization of the suicide attempt group remains operational, potentially resulting in some degree of misclassification.

1. Beck AT, Beck R, Kovacs M. Classification of suicidal behaviors: I. Quantifying intent and medical lethality. The American journal of psychiatry. 1975;132(3):285-7.

2. Kumar CS, Mohan R, Ranjith G, Chandrasekaran R. Characteristics of high intent suicide attempters admitted to a general hospital. Journal of affective disorders. 2006;91(1):77-81.

3. Horesh N, Levi Y, Apter A. Medically serious versus non-serious suicide attempts: Relationships of lethality and intent to clinical and interpersonal characteristics. Journal of Affective Disorders. 2012;136(3):286-93.

4. Haw C, Hawton K, Houston K, Townsend E. Correlates of relative lethality and suicidal intent among deliberate self-harm patients. Suicide and Life-Threatening Behavior. 2003;33(4):353-64.

5. Chae H-R, Jeong J-W, Han S-H, Lee S-Y, Yoon H-D, editors. The Current status of attempting suicide using NEDIS registration data and follow-up management. Proceedings of the Korea Contents Association Conference; 2014: The Korea Contents Association.

---

## [Editor Report · Decision Letter 1]

21 Feb 2024

Risk factors of reattempt among suicide attempters in South Korea: A nationwide retrospective cohort study

PONE-D-23-32199R1

Dear Dr. Ahn,

We’re pleased to inform you that your manuscript has been judged scientifically suitable for publication and will be formally accepted for publication once it meets all outstanding technical requirements.

Kind regards,

Xenia Gonda

Academic Editor

PLOS ONE
---

## [Editor Report · Acceptance letter]

28 Mar 2024

PONE-D-23-32199R1 

PLOS ONE

Dear Dr. Ahn, 

I'm pleased to inform you that your manuscript has been deemed suitable for publication in PLOS ONE. Congratulations! Your manuscript is now being handed over to our production team.

Kind regards, 

on behalf of

Dr. Xenia Gonda 

Academic Editor

PLOS ONE